# An Assessment of the Mechanophysical and Hormonal Impact on Human Endometrial Epithelium Mechanics and Receptivity

**DOI:** 10.3390/ijms25073726

**Published:** 2024-03-27

**Authors:** Anna K. Sternberg, Liubov Izmaylova, Volker U. Buck, Irmgard Classen-Linke, Rudolf E. Leube

**Affiliations:** Institute of Molecular and Cellular Anatomy, RWTH Aachen University, Wendlingweg 2, 52074 Aachen, Germany; asternberg@ukaachen.de (A.K.S.); lizmayolova@ukaachen.de (L.I.); vbuck@ukaachen.de (V.U.B.); iclassen-linke@ukaachen.de (I.C.-L.)

**Keywords:** human endometrial epithelium, menstrual cycle, window of implantation, Ishikawa cells, trophoblast adhesion, stiffness, mechanosensitivity, extracellular matrix, nanoindentation, traction force microscopy

## Abstract

The endometrial epithelium and underlying stroma undergo profound changes to support and limit embryo adhesion and invasion, which occur in the secretory phase of the menstrual cycle during the window of implantation. This coincides with a peak in progesterone and estradiol production. We hypothesized that the interplay between hormone-induced changes in the mechanical properties of the endometrial epithelium and stroma supports this process. To study it, we used hormone-responsive endometrial adenocarcinoma-derived Ishikawa cells growing on substrates of different stiffness. We showed that Ishikawa monolayers on soft substrates are more tightly clustered and uniform than on stiff substrates. Probing for mechanical alterations, we found accelerated stress–relaxation after apical nanoindentation in hormone-stimulated monolayers on stiff substrates. Traction force microscopy furthermore revealed an increased number of foci with high traction in the presence of estradiol and progesterone on soft substrates. The detection of single cells and small cell clusters positive for the intermediate filament protein vimentin and the progesterone receptor further underscored monolayer heterogeneity. Finally, adhesion assays with trophoblast-derived AC-1M-88 spheroids were used to examine the effects of substrate stiffness and steroid hormones on endometrial receptivity. We conclude that the extracellular matrix and hormones act together to determine mechanical properties and, ultimately, embryo implantation.

## 1. Introduction

The human menstrual cycle is unusual, even among mammals. The endometrium undergoes cyclic changes under the influence of ovarian steroid hormones to become receptive to embryo implantation. During the estradiol-driven proliferative phase, the functionalis layer of the endometrium builds up and further differentiates during the early secretory phase that is mainly regulated by progesterone. It prepares the endometrial epithelium to permit embryonic trophoblast adhesion and invasion during the window of implantation. If implantation does not take place, the steroid–hormone concentration drops rapidly, eventually resulting in the shedding of the functionalis. Endometrial epithelial cells undergo a partial epithelial-to-mesenchymal transition (EMT) during the early secretory phase by losing their apico-basal polarity [1,2]. Hormone-induced downregulation of the apical polarity complexes Par and Crumbs [3] results in the rearrangement of the tripartite junctional complex, with keratin intermediate filament-associated desmosomes and actin filament-associated adherens junctions distributed along the entire lateral plasma membrane [4]. In a previous study, it was shown that integrin α6, which is part of extracellular matrix adhesions, also redistributes along the lateral plasma membranes of endometrial epithelial cells under the influence of progesterone [5]. It was suggested that the loss of endometrial epithelial cell polarity is functionally linked to trophoblast adhesion and invasion [2,6,7].

Besides hormone stimulation, endometrial epithelial cell receptivity may also be determined by the mechanical environment [2,8,9]. To date, however, few studies have addressed this aspect. The pioneering study of Jiang et al. [10] applied in vivo magnetic resonance elastography, finding menstrual cycle-dependent changes in endometrial stiffness. The resolution of the method, however, was quite limited and did not allow for distinguishing between the different endometrial subcompartments. 

Atomic force microscopy has proven to be an excellent tool to assess the mechanical properties of soft matter. It has been used to determine the stiffness of biological samples by the nanoindentation of tissues and cells [11]. Furthermore, the method allowed for an investigation of the effects of mechanical cues on cell behavior and characterization of the mechanisms of mechanotransduction and mechanosensing in vital systems mimicking changes in force balances induced by internal and external forces [12]. Nanoindentation has, therefore, become a versatile tool to apply defined forces at cellular resolution and untangle complex relationships of bio-mechanical interactions in living matter. The recent publication of Abbas et al. [13] investigated uterine tissue samples ex vivo by atomic force microscopy. They found changes in the stiffness of the decidua, presumably caused by trophoblast invasion during pregnancy, but did not analyze menstrual cycle-dependent changes in endometrial stiffness [10,13].

In the current study, we wanted to explore endometrial epithelial cell mechanics in different mechanophysical environments and hormonal conditions that may be relevant to trophoblast attachment during the window of implantation. We hypothesized that extracellular matrix- and hormone-induced epithelial cell remodeling goes along with alterations in cellular stiffness, traction, and stress response, which work together to support embryo adhesion and implantation. To study this complex cross-talk, we decided to use the well-established, hormone-sensitive, human endometrial epithelial adenocarcinoma cell line Ishikawa [14]. Ishikawa cells were seeded on top of hydrogels with a defined stiffness to mimic the stromal–epithelial interface. Using adhesion assays with trophoblast-derived spheroids, we aimed to examine the effects of mechanical and hormonal stimulation on endometrial receptivity.

## 2. Results

### 2.1. Endometrial Epithelial Ishikawa Cells Are Sensitive to Substrate Stiffness

Given that endometrial epithelial cells experience different levels of stiffness of underlying substrates during the menstrual cycle because of stromal compartment decidualization, we placed hormone-sensitive Ishikawa cells [14] on polyacrylamide-based hydrogels with a defined stiffness. Polyacrylamide gels are commonly used as cell culture substrates to mimic the mechanics of the naturally occurring soft extracellular matrix. They are homogeneous, comparatively easy to handle, allow for precise stiffness tuning, and are robust with respect to batch-to-batch variation. In contrast to other routinely used soft cell substrates such as polydimethylsiloxane, cells grown on top of polyacrylamide gels do not deposit an extracellular matrix onto the gel. Therefore, the biomechanical and biochemical composition of polyacrylamide-based matrices can be separately controlled (e.g., [15,16]). The chosen stiffness range was guided by the reported Young’s moduli measured by in vivo magnetic resonance elastography of human uteri [10]. Based on these measurements, we chose 4 kPa to model the extracellular matrix of the healthy endometrium. In pathological states such as preeclampsia, endometrial extracellular matrix stiffness increases to 19–26 kPa [17,18]. We therefore used a substrate stiffness of 23 kPa to mimic a supra-physiological condition. The softest substrate on which Ishikawa cells formed proliferating colonies was 1 kPa. We thus used this stiffness to imitate a sub-physiological condition. The hydrogel surfaces were functionalized with growth factor-reduced Matrigel to provide a substrate consisting of physiologic basement membrane components to support the adhesion and growth of the Ishikawa cells (Figure 1A).

When the Ishikawa cells were placed on soft polyacrylamide gels with 1 kPa stiffness, they were not able to form contiguous monolayers but, instead, formed islands consisting of cells that locally stratified and generated peripheral, finger-like protrusions (Figure 1B). The same phenotype was occasionally observed on 4 kPa polyacrylamide gels. But, in most instances, complete monolayers were formed under these conditions, as was consistently the case for cells growing on 23 kPa hydrogels and on glass (Figure 1B) or plastic (Appendix A).

We noted that cell morphology differed between the different conditions. To measure the cell area, monolayers were first stained with antibodies against the tight junction marker ZO1 to delineate the cell borders by immunofluorescence microscopy. The recorded fluorescence images were segmented and used for digital analyses. The analyses revealed that the cell area correlated with matrix stiffness. The softer the substrate was, the smaller the cells were, leading to an increased number of cells per area (Figure 1C,D).

In the next set of experiments, we examined cell height by staining for filamentous actin, which is enriched in the cell cortex underneath the plasma membrane (Figure 1E). Significant differences in cell height were readily apparent between monolayers on glass versus monolayers on polyacrylamide gels (Figure 1F). A slight, though not significant, reduction in cell height was noted in monolayers on 23 kPa versus 1 kPa and 4 kPa hydrogels (Figure 1F).

### 2.2. Substrate Stiffness Regulates the Response of Ishikawa Cells to Constant Strain

It has been proposed that the topology and morphology of epithelial monolayers reflect their mechanical properties [19,20,21], suggesting that the observed substrate-dependent shape changes in Ishikawa monolayers are a consequence of altered cell mechanics. In the next set of experiments, we therefore studied the mechanical properties of Ishikawa cell monolayers growing on hydrogels with different levels of stiffness by nanoindentation. This setup simulates the reported pushing of the attached blastocysts [22] onto the underlying endometrial epithelium at single-cell resolution. A spherical probe was pressed into the apical cell domain up to a defined depth (Figure 2A). The force needed to indent the cell provided a measure for cellular stiffness, i.e., the effective Young’s modulus (Figure 2B). We used a maximum indentation depth of 2 µm, which corresponded to ~10% of the total cell height (see Figure 1F) for all experiments, and applied the Hertzian contact model only to the first 1 µm of the indentation curve. The measurements did not detect a difference in the effective Young’s modulus for Ishikawa cells growing on 1 kPa, 4 kPa, or 23 kPa polyacrylamide substrates (Figure 2C).

Given the substrate stiffness-independent cellular viscoelasticity, we wondered whether hormonal stimulation elicits an endometrial epithelial stiffness response. We first confirmed the overall hormone responsiveness of our Ishikawa cells by immunoblot detection of the estradiol-dependent induction of progesterone receptors A and B (Appendix A; [14,23,24,25]). Combined estradiol and progesterone treatment resulted mainly in the induction of progesterone receptor B. This was in accordance with the in vivo situation [26,27]. Nanoindentation experiments of hormone-treated Ishikawa cells growing on substrates of different stiffness, however, did not reveal any differences in the effective Young’s modulus (Figure 2C).

To investigate the dynamics of acute mechanical responses in Ishikawa monolayers, we performed relaxation experiments. To this end, we held the probe at an indentation depth of 2 µm for 30 s and measured the mechanical relaxation time. As expected for viscoelastic materials, the cells showed non-linear stress–relaxation behavior in all experimental setups. Yet, cells growing on 1 kPa polyacrylamide gels relaxed more slowly than Ishikawa cells growing on stiffer 4 kPa and 23 kPa substrates. The effect was most pronounced for estradiol-stimulated Ishikawa cells (Figure 2D). These findings indicated reduced mechanical responsiveness of Ishikawa cells growing on soft substrates, which is in line with the reduced ability to form contiguous monolayer sheets, and may be of relevance to endometrial receptiveness.

### 2.3. Progesterone/Estradiol Treatment Amplifies Traction Force Hotspots in Confluent Ishikawa Cell Monolayers

Traction force microscopy was performed next to examine changes in global cell mechanics within the Ishikawa monolayers. With this method, the displacement of the fluorescent fiducial beads that were incorporated in the polyacrylamide gels with a defined stiffness was measured after the removal of the monolayer (Figure 3A). From this, the forces exerted by the epithelium onto the substrate could be calculated at high spatial resolution. Then, 4 kPa and 23 kPa hydrogels were compared regarding physiological and supraphysiological stiffness. Cells grown on the stiffer gel showed significantly increased mean traction forces (92.09 ± 21.11 Pa versus 248.15 ± 109.90 Pa; Figure 3B). Hormone treatment did not change the mean traction forces in the different stiffness environments (Figure 3B). Next, heat maps were prepared to depict the traction force patterns. They revealed that traction forces were homogenously distributed throughout the entire Ishikawa monolayers in the absence of hormones and in the presence of estradiol (example at left in Figure 3C). However, when cells were stimulated with a combination of estradiol and progesterone, we observed an increased number of force hotspots (Figure 3C, right). The local forces reached peak values of 1000 Pa, which were surrounded by large areas of low traction (50–100 Pa). Traction hotspots were only detected for cells on a soft gel with physiological stiffness (4 kPa). The quantification of the number of force hotspots per area and the distance between force hotspots uncovered highly significant differences between estradiol- and estradiol/progesterone-treated Ishikawa cell monolayers grown on physiological 4 kPa substrates (Figure 3D,E). These observations demonstrate that the addition of progesterone induces locally restricted alterations in endometrial epithelial cell mechanics.

### 2.4. Ishikawa Monolayers Show Heterogeneity in Progesterone Receptor and Vimentin Expression

Next, we wanted to find out whether the observed traction force hotspots were mirrored by the heterogeneity of the protein expression in the endometrial epithelial monolayer in the presence of progesterone. Based on the known hormone-dependent upregulation of the progesterone receptor (Appendix A; [25]), we performed immunostaining to examine its induction at the cellular level. The staining of confluent Ishikawa cells growing on 4 kPa polyacrylamide hydrogels in the presence of both steroid hormones revealed that 10–20% of cells were positive. As expected, hardly any progesterone receptor-positive cells could be detected in non-stimulated monolayers. Furthermore, staining intensity and the number of positive cells were slightly higher in estradiol- than in estradiol/progesterone-treated cells. Remarkably, the progesterone receptor immunosignal was irregularly distributed within the monolayer in each condition (Figure 4). We therefore concluded that only a small population within the monolayer could respond directly to progesterone treatment.

It has been suggested that endometrial epithelial cells undergo partial epithelial-to-mesenchymal transition during the secretory phase of the menstrual cycle to support embryo adhesion and invasion [1,2,28]. We therefore stained the hormone-treated Ishikawa cells with antibodies against the mesenchymal marker and cytoskeletal intermediate filament polypeptide vimentin. Positive single cells and small cell clusters were detected within the monolayer (Figure 4). We could, however, not detect differences in the number of positive cells, which ranged between 1 and 5% in all experimental groups irrespective of the hormone treatment. We also found no direct spatial correlation between progesterone receptor- and vimentin-positive cells (Figure 4). Together with the traction force hotspots, however, we can conclude that the Ishikawa monolayers were highly heterogeneous, providing spatially restricted mechanically and biochemically defined microenvironments.

### 2.5. Substrate Stiffness Scales with Trophoblast Adhesion of Ishikawa Cells

To find out how substrate stiffness and hormone-dependent heterogeneity cooperate to modulate endometrial receptivity, we employed a confrontation assay in which we placed trophoblast-derived choriocarcinoma/hybridoma AC-1M-88 spheroids on Ishikawa monolayers. To standardize the assay, we grew spheroids in a scaffold-free system (Figure 5A). Dissociated AC-1M-88 cells were seeded into 300 µm diameter agarose-based micromolds and grown for 3 days. Cells aggregated during this time and formed compact spheroids (“trophospheres”) of 160 ± 15 µm in size. (Figure 5B,C). Twenty spheroids were individually placed with a micropipette at regular spacing on top of Ishikawa cell monolayers (Figure 5D). After incubation for two hours, non-adherent spheroids were flushed off by repeated rigorous washing of the monolayers with PBS. The remaining attached spheroids were counted. Using this assay, we found that trophospheres attached more firmly to Ishikawa monolayers growing on stiff matrices. This effect was observed in the non-stimulated and hormone-treated Ishikawa cells (Figure 5E). Trophosphere adhesion was highest in monolayers treated only with estradiol (Figure 5F).

## 3. Discussion

In this study, we investigated the interplay between mechanics, steroid hormone stimulation, and trophoblast receptivity in endometrial epithelial Ishikawa monolayers.

### 3.1. Substrate Stiffness Alters Ishikawa Cell Mechanics

To study the effect of substrate stiffness on the hormone-responsive Ishikawa cells, we first established a two-dimensional in vitro model that resembled the luminal endometrial epithelium by seeding the Ishikawa cells on top of Matrigel-functionalized polyacrylamide gels. This setup differed from most previous 2D in vitro studies (e.g., [29,30,31]), which grew cells on hard, i.e., non-physiological, surfaces such as glass or plastic. According to the manufacturers, they had a stiffness of ~70 GPa and ~3–4 GPa (see also Miyake et al., 2006 [32]), respectively, which was in stark contrast to the 0.3–4 kPa most likely encountered in vivo [10,13,33]. We found that substrate stiffness impacted cell morphology and monolayer formation ability. These findings are in agreement with reports on primary, patient-derived endometrial epithelial cells, which have been shown to exhibit substrate stiffness-dependent differences in cell morphology and actin fiber organization [34].

The observed substrate stiffness-dependent changes in cell morphology can be taken as a strong indication of altered monolayer mechanics [19,20,21]. Experimental evidence for this conclusion was provided by the detection of substrate stiffness-dependent differences in stress–relaxation behavior in response to the nanoindentation of cells in the Ishikawa monolayer. The observed responses were likely linked to differences in the cortical actomyosin system [35]. We suggest that the indentation of the spherical nanoindenter tip mimics the force exerted by the attached blastocyst on the endometrial epithelium. It likely elicits an actomyosin-dependent response, which may be directly coupled to the remodeling of cell–cell junctions and cell–basement membrane junctions [4,36].

### 3.2. Combined Treatment with Estradiol and Progesterone Induces Traction Force Hotspots

To better understand the implications of cytoskeletal remodeling on endometrial epithelial cell mechanics, traction force analyses were performed. Traction forces are transmitted to the basement membrane via integrins and reflect the intracellular tension of the actomyosin network [37]. To our surprise, combined treatment with estradiol and progesterone resulted in the appearance of traction force hotspots. These hotspots comprised small foci, where traction forces were about ten times higher than in the neighboring areas. Hence, we concluded that only a limited population of cells within the endometrial epithelial cell monolayer responded to progesterone treatment and changed their cytoskeletal architecture, leading to local alterations of the intercellular stress balance. It was, therefore, of interest to find out that the progesterone receptor and vimentin were only detectable in single cells and small cell clusters spread throughout the monolayer. In support, steroid hormone receptors were non-homogenously distributed in endometrial epithelial cells and showed topographic differences between the uterine fundus and cervix [38,39]. The same held true for vimentin [40]. Our results, however, do not support the simple scenario in which progesterone induces local responses that are directly linked to vimentin expression and increased cellular traction.

### 3.3. The Trophoblast Receptivity of Ishikawa Cells Is Affected by Mechanical and Hormonal Cues

To test the functional properties of endometrial epithelial cells under different hormonal and mechanical conditions, we confronted the 2D Ishikawa monolayers with trophoblast-derived trophospheres. Most remarkably, increasing trophoblast receptivity scaled with substrate stiffness. Stiff substrates supported trophoblast adhesion most efficiently. This observation agrees with another recent study [41] that examined human choriocarcinoma JAR cells growing on polyacrylamide-based stiffness gradients. The cells were more spread out, formed more focal adhesions and actin stress fibers, and migrated faster on stiffer substrates, a behavior that is referred to as durotaxis. Elevated turgor of the decidualized stroma may thus support trophoblast adhesion. How the trophospheres in our multicellular system sensed and responded to the different levels of substrate stiffness remains to be shown. It must involve a mechanosensing mechanism that transduces the information through the Ishikawa monolayer. A likely explanation is that changes in the cytoskeletal and junctional organization of the endometrial cells are involved. Taken together, our results emphasize the importance of accounting for substrate stiffness in in vitro models of implantation.

The observation that trophoblast attachment rates are higher in the presence of estradiol in comparison to those in the presence of both estradiol and progesterone is in contrast to the report of Singh et al. [42]. This study differed from ours by using heterologous murine blastocysts instead of human trophoblasts, by implementing a different hormone treatment protocol (pre-treatment with E2 prior to combined E2 + MPA; only 48 h treatment), by the lack of an extracellular matrix, and by a much longer co-culture time, all of which may have contributed to the apparent discrepancy. Interestingly, Flamini et al. [43] observed that treatment with estradiol alone induced rapid actin and vinculin translocation from the cytoplasm toward the plasma membrane in single, cultured Ishikawa cells. Whether this explains the increase in trophoblast attachment rates in our setup has to be further studied. The not fully understood results indicate that there are still cellular, biochemical, and mechanical factors lacking in our in vitro system, as previously suggested in a comparable setup [31].

## 4. Materials and Methods

### 4.1. Cell Culture

The endometrial adenocarcinoma cell line Ishikawa (ECACC 99040201; RRID: CVCL_2529) and hybridoma extravillous trophoblast cell line AC-1M-88 ([44,45,46]; RRID: CVCL_1803) were kept at 37 °C in a humid atmosphere with 5% CO_2_ in phenol red-free Dulbecco’s modified Eagle’s medium (DMEM)/F12 medium (Sigma-Aldrich, Saint Louis, MO, USA) supplemented with 10% (*v*/*v*) steroid hormone-free fetal calf serum (C.C.Pro, Oberdorla, Germany), 2.5 mM L-glutamine (Gibco, Paisley, UK), and 10% (*v*/*v*) antibiotic-antimycotic solution comprising penicillin, streptomycin, and amphotericin B (Sigma-Aldrich). Both cell lines were passaged before they reached confluence by washing them with phosphate-buffered saline (PBS, Biochrom, Berlin, Germany) containing 0.02% (*v*/*v*) 2,2′,2″,2‴-(ethane-1,2-diyldinitrilo-)tetraacetic acid (EDTA, Sigma-Aldrich) for 3–5 min at 37 °C and incubated in PBS containing 0.25% (*w*/*v*) trypsin (BD Biosciences, San Jose, CA, USA) and 0.25% EDTA for 3 min at 37 °C. Cells were resuspended in culture medium and passaged in a ratio of 1:10 once a week. Experiments were restricted to passages 3–15 after thawing.

For hormone treatment, we substituted progesterone by the more stable analog medroxyprogesterone acetate (MPA) throughout. Stock solutions of E2 and MPA (both from Sigma Aldrich) were prepared in ethanol at 1 × 10^−3^ M and 1 × 10^−2^ M, respectively, and stored at −20° C. Up to two weeks before use, stock solutions were diluted 1:1000 and 1:100 in cell culture medium, and aliquots were immediately frozen at −20 °C. For the preparation of the hormone-containing cell culture medium, aliquots were thawed and further diluted to their respective final concentration in the cell culture medium (10 nM for E2 and 1000 nM for MPA). Hormone-diluent ethanol was added to the medium of the unstimulated control (0.03% *v*/*v*). Of note, hormones and media were freshly mixed for each medium exchange, which was performed daily to take into account the reported metabolization of steroid hormones in Ishikawa cells [23,47].

### 4.2. Hydrogel Preparation and Endometrial Epithelial Cell Monolayer Cultivation

Ishikawa cells were seeded on glass coverslips (diameter 15 mm; Langenbrinck Menzel/Thermo Fisher Scientific Inc., Waltham, MA, USA) or on soft polyacrylamide (PAA) hydrogels (a scheme of the workflow is shown in Appendix A). The glass cover was coated at 4 °C overnight with a solution containing 40 µg mL^−1^ phenol red-free and growth factor-reduced Matrigel (BD Biosciences) in PBS. Glass-bottomed dishes (Cellvis, Sunnyvale, CA, USA) were treated with 0.1 M sodium hydroxide (Merck, Darmstadt, Germany) for 5 min followed by incubation with 4% (*v*/*v*) 3-aminopropyltriethoxysilane in isopropanol (Sigma-Aldrich). After washing with water, the glass surface was activated with 1% (*v*/*v*) glutaraldehyde (Merck) for 30 min to allow for the covalent linking of a PAA gel to the surface. PAA gels were prepared with different concentrations of bis-acrylamide (Carl Roth, Karlsruhe, Germany) and acrylamide (Sigma-Aldrich). Final concentrations were 0.03% bis-acrylamide and 5% acrylamide for gels of 1 kPa stiffness, 0.1% bis-acrylamide and 5% acrylamide for gels of 4 kPa stiffness, and 0.2% bis-acrylamide and 7.5% acrylamide for gels of 23 kPa stiffness. Polymerization was initiated by adding 0.05% N, N, N′, N′-tetramethylethylendiamin (TEMED, Bio-Rad, Hercules, CA, USA) and 0.5% ammonium persulfate (Merck). The Young’s modulus of the gels was confirmed by nanoindentation using a Chiaro Nanoindenter (Optics11 Life, Amsterdam, Netherlands). Measurements were performed for each batch in triplicate (Appendix A). To covalently bind the extracellular matrix proteins to the gel surface, a previous protocol was adapted [48]. Briefly, the gels were functionalized by 0.3 µg mL^−1^ N-hydroxysuccinimide ester (Sigma-Aldrich) in 0.05 M 4-(2-hydroxyethyl)-1-piperazineethanesulfonic acid buffer (HEPES, Merck) containing 2 µL mL^−1^ tetramethacrylate (Sigma-Aldrich) and 25 µg mL^−1^ 2-hydroxy-4′-(2-hydroxyethoxy)-2-methylpropiophenone (Sigma-Aldrich). After adding 500 µL of a functionalizing solution, the gel was cross-linked under ultraviolet light with a wavelength of 253 nm for 10 min (Uvo-Cleaner 42-220, Jelight Company, Irvine, CA, USA). The gels were immediately transferred on ice and washed 2× with 25 mM HEPES buffer for 5 min and 2× with PBS for 5 min. A total of 100 µL of 40 µg mL^−1^ phenol red-free growth factor-reduced Matrigel solution was added on top of the gels and incubated overnight at 4 °C. The gels were rinsed 3× with sterile PBS, sterilized under UV light for 30 min, and equilibrated in a cell culture medium for 15 min. Cells were seeded at a final density of 1 × 10^6^ cells per culture dish. Cells were treated with either 17β-estradiol (E2) or a combination of 17β-estradiol and medroxyprogesterone acetate (MPA, Sigma-Aldrich) for 5 days and the medium was changed every day. The final concentrations were 1 × 10^−8^ M for E2 and 1 × 10^−6^ M for MPA. Monolayers were cultivated for 5 days.

### 4.3. Growth of Trophoblast Spheroids

Multicellular trophoblast spheroids were grown in agarose micromolds that were prepared by pouring 3% (*w*/*v*) agarose (Biozym Scientific GmbH, Hessisch Olendorf, Germany) dissolved in distilled sterile H_2_O (B. Braun, Melsungen, Germany) into silicone stencils (#12-256, MicroTissues Inc., Sigma-Aldrich) and sterilized under UV light for 30 min. Molds were transferred into a 12-well plate (CytoOne, StarLab International GmbH, Hamburg, Germany) and equilibrated in culture medium for 30 min. Trypsinized AC-1M-88 cells were suspended at 1,000,000 cells/mL, seeded into the agarose molds (200 µL/mold), and allowed to settle for 10 min at 37 °C. Additional culture medium was added to the wells. Growth and spheroid formation were monitored on a Zeiss Axiovert 135 Inverted Fluorescence Phase microscope (Carl Zeiss Microscopy, Jena, Germany) for 72 h.

### 4.4. Confrontation of Trophoblast Spheroids with Ishikawa Monolayers

Ishikawa monolayers were grown for 5 days on plastic or PAA gels of 1 or 4 kPa stiffness. Immediately before the confrontation experiment, information about gel stiffness and hormonal treatment was blinded by an uninvolved lab member. To do so, dishes were renamed by a random number that was created using the “=RANDBETWEEN(0;100)” function of Microsoft Excel (Microsoft, Redmond, WA, USA). A total of 25 mM of HEPES was added to the culture medium and cell culture dishes were placed on top of a heating plate (Störk-Tronic, Stuttgart, Germany) set to 37 °C. Twenty AC-1M-88 spheroids were carefully placed on top of each Ishikawa monolayer using an EZ-Grip pipette (CooperSurgical, Trumbuli, CT, USA), without moving the dishes. After 2 h, the monolayers were washed 3× with 1 mL PBS. The direction of pipetting was strictly controlled and uniform for each dish. The PBS jet was always directed toward the edge of the dish and never directly at spheroids. The number of the remaining attached spheroids was then counted.

### 4.5. Immunofluorescence

For antibody staining, cells were fixed with ice-cold methanol (Carl Roth) for 5 min at −20 °C, air dried for 15 min at room temperature, and rehydrated in PBS for 5 min. Detailed information on antibodies and controls is provided in Appendix A. Blocking reagents and antibodies were diluted in PBS supplemented with 1.5% bovine serum albumin (*w*/*v*, Serva, Heidelberg, Germany). Cells on coverslips were incubated with primary and secondary antibodies for 45 min each. For the staining of cells on polyacrylamide gels, incubation times were prolonged to 24–72 h. Primary antibodies were incubated overnight at 4 °C and secondary antibodies were incubated in the dark at room temperature or at 4 °C. For nuclear staining, 1 µg/mL Hoechst 33342 (Sigma-Aldrich) was added to the staining solution of the secondary antibody. After each incubation step, cells were washed with PBS 3× for 10 min. Stained structures were briefly rinsed with H_2_O and mounted with Mowiol 4-88 (Sigma-Aldrich). For the staining of F-actin, cells were fixed with 4% paraformaldehyde (Carl Roth) in PBS for 20 min at room temperature. Cells were washed 3× in PBS and permeabilized for 3 min with 0.1% Triton X-100 (Sigma-Aldrich) in PBS. F-actin was labeled by overnight incubation with phalloidin that had been tagged with Alexa 488 (Thermo Fisher Scientific) and diluted 1:200 in PBS.

### 4.6. Microscopy and Image Processing

Bright-field images were obtained with the help of a Zeiss Axiovert 135 microscope (Carl Zeiss Microscopy) equipped with a standard color camera (Canon 650D, Canon, Ota, Japan) and a low magnification objective (Leitz LD A-Plan 20×/0.3, Wetzlar, Germany). Fluorescence stainings were imaged by structured illumination microscopy using an ApoTome.2 mounted onto an Axio Imager M.2 microscope using an Axiocam MRm camera (Carl Zeiss Microscopy) via a 20× objective (Plan-Apochromat 20×/0.8 DIC, Carl Zeiss Microscopy) or a 63× oil objective (63×/1.4 Oil DIC M27, Carl Zeiss Microscopy) with ZEN 3.0 blue edition software (2464 × 2056 pixel, binning 2X2). For the recording of actin- and ZO1-staining in AC-1M-88 spheroids and Ishikawa monolayers, a Zeiss LSM 710 DUO confocal microscope was used. For spheroid visualization, single spheroids were transferred in PBS into a glass-bottomed dish (14 mm diameter, thickness no. 1·5, MatTek, Ashland, MA, USA) and imaged with an LGK 7872 ML8 argon ion laser with appropriate emission bandpass filters via a 20× objective (20×/0.8 M27) using Zen black 2.1 SP3 software (all Carl Zeiss Microscopy). Ishikawa cells were visualized using the same setup but with a 40× objective (LD C-Apochromat 40×/1.1 W Korr M27, Carl Zeiss Microscopy). Image processing was performed with the open-source, image-processing program Fiji based on ImageJ [49].

### 4.7. Cell Area Analysis

Cell borders were detected by immunostaining for the tight junction marker ZO1 in Ishikawa cells grown on substrates with different levels of stiffness (1, 4, 23 kPa and glass). Confocal fluorescence recordings were processed as maximum intensity projections in Fiji software version 1.54f. The images were then segmented using the anatomical segmentation algorithm cellpose version 2.2.1 ([50]; www.cellpose.org). Cells at the borders of the images were discarded to avoid segmentation errors. Cell outlines were saved as .txt files. Cell area was then calculated in Fiji based on ImageJ using the imagej_roi_converter.py plugin.

### 4.8. Immunoblotting

For immunoblotting, Ishikawa cells were seeded on 100 mm plastic cell culture dishes. Cells were treated with either E2 or the combination of E2 and MPA for 5 days as described above and the medium was changed every second day. For protein extraction, monolayers were rinsed in PBS and put on ice. Ice-cold cell lysis buffer containing 50 mM Tris (Biomol, Hamburg, Germany), 1% (*v*/*v*) Nonidet P40 (Sigma-Aldrich), and 150 mM sodium chloride (AppliChem, Darmstadt, Germany) supplemented with complete protease inhibitor (Roche, Mannheim, Germany) was added and cells were removed from their cell culture dish by scraping with a cell scraper (VWR International, Radnor, PA, USA). The medium containing the detached cells was transferred to a new 1.5 mL Eppendorf tube, vortexed, and put on a shaker at 4 °C for 30 min. The cell lysate was centrifuged at 12,000 rpm at 4 °C for 20 min. The supernatant was stored at −20 °C. For immunoblotting, Laemmli buffer containing 100 mM Tris-HCl (pH 8; Gibco), 10% (*v*/*v*) glycerol (Merck), 3% (*w*/*v*) sodium dodecyl sulfate (SDS, Carl Roth), 7.5% (*v*/*v*) β-mercaptoethanol (Sigma-Aldrich), and 250 µg mL^−1^ bromophenol blue (Sigma-Aldrich) was added to the protein lysate. Proteins were denatured at 37 °C for 30 min and separated by electrophoresis through a 10% SDS-polyacrylamide gel in a buffer containing 25 mM TRIS-HCl (pH 6.8), 250 mM glycine (Biomol), and 0.01% SDS. The separated proteins were electroblotted from the gels onto polyvinylidene difluoride (PVDF) membranes (Merck Millipore Ltd., Cork, Ireland) in a buffer containing 25 mM TRIS, 192 mM glycine, and 10% ethanol at 100 V for 1 h. The membrane was blocked in PBS-T (PBS containing 0.05% Tween-20, Sigma-Aldrich) with 5% low-fat milk powder for 2 h at room temperature and incubated with a primary antibody diluted in PBS-T with 2% low-fat milk powder overnight at 4 °C. Antibody dilutions were 1:1000 for the monoclonal mouse anti-progesterone receptor (clone PgR636; M3569, DAKO, Jena, Germany) and monoclonal rabbit anti-estrogen receptor (clone Sp1; RM 9101-SO, Thermo Fisher Scientific) and 1:5 000 for the polyclonal rabbit anti-actin antibody (A2066, Sigma-Aldrich). After washing with PBS-T, membranes were incubated for 1 h with the horseradish peroxidase-conjugated secondary antibody diluted 1:5 000 in PBS-T with 2% low-fat milk powder (anti-mouse IgG and anti-rabbit IgG; DAKO). The detection was performed using Pierce ECL Western Blotting Substrate (Thermo Fisher Scientific), and chemiluminescence signals were captured using the FusionSolo imaging system (Fusion SL, Vilber Lourmat, Marne la Vallée, France). To remove antibodies from the membrane, it was incubated in a stripping buffer containing 0.1 M glycine (pH 2) 3× for 20 min.

### 4.9. Nanoindentation

The elastic properties of the endometrial epithelial cell monolayers were measured using a Chiaro Nanoindenter equipped with an OP1550 interferometer (Optics11 Life) that was mounted on a Zeiss Axio Observer seven microscope (Carl Zeiss Microscopy). Experiments were performed at 37 °C in an incubation chamber and HEPES was added to the culture medium (final concentration 25 mM). The spherical probe tips had a radius of 8.5–10 µm and the cantilever spring constant was between 0.41 and 0.49 N/m. Optical and geometrical calibration was performed according to the recommendations of the manufacturer before every experiment. Cells and probe tips were placed in the incubation chamber at least 30 min prior to taking measurements to avoid measurement errors due to temperature drift. To keep experimental parameters as standardized as possible [51], we performed the measurements with similar cell passages, identical numbers of cells for seeding, comparable confluences, identical cell culture media, identical surface coatings, and the same individual handling the cells. Single cells in confluent monolayers were measured by placing the probe tip above the cell center. The surface was found using the implemented “Find Surface” function. The indentation profile was applied by indenting 2 µm into the cell surface within 2 s. Data were evaluated using the DataViewer V2.5 software (Optics11 Life). To determine Young’s modulus E_eff_, the Hertzian contact model was applied to a maximum indentation depth of 1 µm [52]. Stress–relaxation was measured via the application of a constant strain by holding the tip at a depth of 2 µm for 45 s. Recorded response load over time was analyzed using an open-source Jupyter notebook [53]. Briefly, the loading curves of a measurement series were aligned, and the maximum force and corresponding time were assigned as time point 0. The average curve was plotted and the relaxation time at which the stress response relaxed to 70% of its original value was calculated by the software [54].

### 4.10. Traction Force Microscopy

Traction force microscopy was performed on an inverted Zeiss Axio Observer 7 microscope via a 20× objective (20×/0.8 M27) using Zen 2.6 pro software (all Carl Zeiss Microscopy) on confluent endometrial epithelial cell monolayers. PAA gels contained 6.5% (*v*/*v*) 0.5 µm fluorescent carboxylated polystyrene beads (Sigma-Aldrich) and were polymerized upside down to let them sink toward the surface. All experiments were performed at 37 °C in an incubation chamber and HEPES was added to the medium (final concentration 25 mM). Images of the reference beads were acquired. To obtain a reference image of the substrate in a relaxed state, 0.1% SDS was added to the medium, and cell detachment was observed under optical control. Images of stressed and relaxed bead positions were aligned by using the open-source image-processing program Fiji based on ImageJ and particle image velocimetry (PIV) was performed using the PIV plugin [55]. Fourier-transform traction cytometry (FTTC, pixel size: 0.4478 µm, Poisson ratio: 0.5) was done using the FTTC plugin [55]. Resulting traction force maps were post-processed in MATLAB (MathWorks, Portola Valley, CA, USA).

### 4.11. Statistical Analysis

Each experiment included at least three biological replicates (N), each consisting of a variable number of technical replicates (n), as specified in the figure legends. Statistics were performed using GraphPad Prism 9.3.1 (GraphPad Software Inc., San Diego, CA, USA). Outlier testing was performed by Grubb’s test. All data were tested for normality by the Shapiro–Wilk test and for homoscedasticity by the Brown–Forsythe test. Parametric data were analyzed by Student’s *t*-test and one-way or two-way ANOVA with Bonferroni’s post-hoc test; non-parametric data were analyzed by the Mann–Whitney U test, Kruskal–Wallis test, or Welsh’s ANOVA. Parametric data were presented as mean ± standard deviation (SD), non-parametric as median with interquartile range and variation. *p* values < 0.05 were considered to be significant.

## 5. Conclusions

The established setup allowed for dissection and analysis of the effects of defined cues, i.e., extracellular matrix stiffness and hormonal stimulation, on endometrial epithelium properties and trophoblast adhesion. The limitations of our setup are the use of transformed/immortalized cell lines, the use of a non-physiological extracellular matrix, and the lack of stromal cells. With these limitations in mind, we were able to show that endocrine cues provided by steroid hormones and the mechanical environment influence endometrial epithelial properties and, most importantly, endometrial function, i.e., receptivity. Moreover, our study indicates that the human endometrial epithelium must be viewed as a highly complex tissue [56,57,58], with heterogeneities that respond differently to environmental cues [26,38,39].

We propose that the induced remodeling of endometrial epithelial cells during the window of implantation occurs in patterns that might be linked to the intrinsic intraepithelial heterogeneity and the biomechanical environment. To look for a strategy to treat endometriosis or endometrial cancer, it is important to look not only for hormonal suppressive therapy but also to interrupt the mechanical interactions between endometrial cells and the pathologically changed abnormal extracellular matrix.

## Figures and Tables

**Figure 1 ijms-25-03726-f001:**
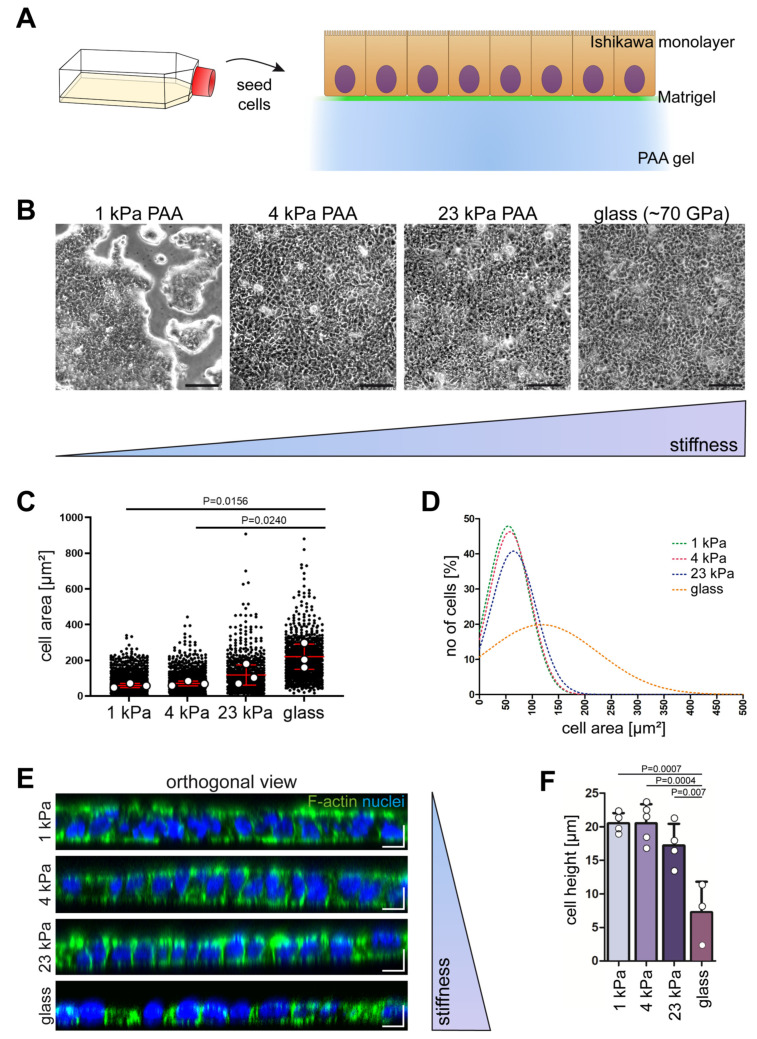
Monolayer formation capacity and the morphology of Ishikawa cells depend on substrate stiffness. (**A**) Schematic of the experimental setup showing that Ishikawa cells were seeded on top of polyacrylamide (PAA) hydrogels with different levels of stiffness that were coated with growth factor-reduced Matrigel. (**B**) Phase contrast images of Ishikawa cells grown on 1 kPa, 4 kPa, and 23 kPa hydrogels and on Matrigel-coated glass. Note that confluent monolayers were only obtained on hydrogels stiffer than 1 kPa. Scale bars = 100 µm. (**C**) The plot depicts the cell area determined on substrates with different levels of stiffness. Each data point represents the area of a single cell (1 kPa: 1658 cells; 4 kPa: 1249 cells; 23 kPa: 869 cells; glass: 1040 cells). The red line represents the mean. Data were obtained by at least 3 independent experiments and were analyzed by one-way ANOVA with Bonferroni post-hoc test. Note that significant increases in cell area correlated with substrate stiffness. (**D**) The graph shows the Gaussian fit of the cell area distribution of Ishikawa cells from (**C**), revealing a much broader cell size distribution of the Ishikawa cells grown on glass in comparison to those grown on soft hydrogels. (**E**) The orthogonal projections show phalloidin-stained F-actin (green) and Hoechst 33342-labeled nuclei (blue) in Ishikawa cells grown on soft polyacrylamide gels of different levels of stiffness and glass for 5 days. Scale bars = 10 µm. (**F**) The bar plot shows the quantification of the determined cell height from phalloidin-stained Ishikawa monolayers on different substrates. The data were obtained by at least 3 independent experiments (n = 20 cells per condition; mean ± SD; one-way ANOVA with Bonferroni post-hoc test).

**Figure 2 ijms-25-03726-f002:**
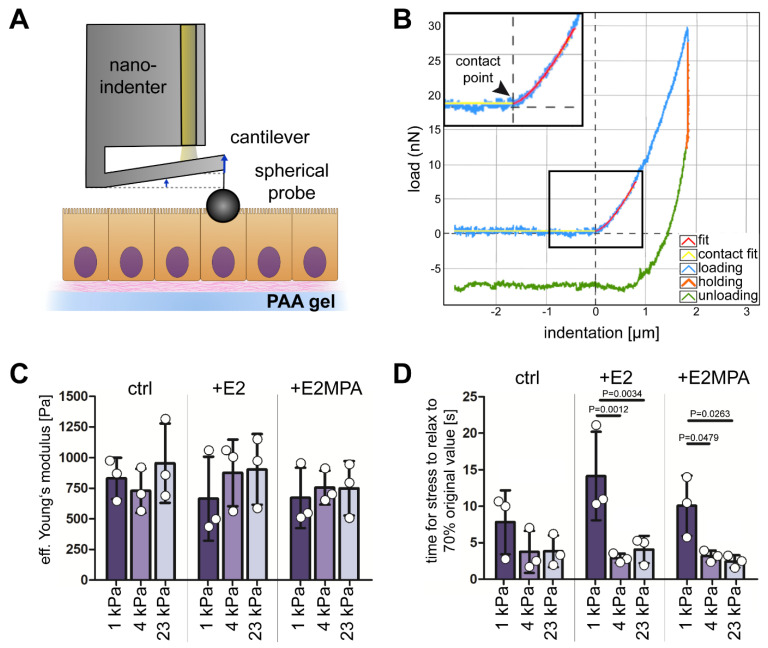
The stress–relaxation behavior of the apical cell domain in Ishikawa cells depends on matrix stiffness. (**A**) A schematic illustration of the principle of nanoindentation. A spherical glass probe of the nanoindenter is indented into a cell monolayer to a defined depth using an optical fiber sensor. Cantilever bending enables calculation of the force that is needed for probe indentation. (**B**) A typical load–displacement curve obtained by nanoindentation measurement of a cell. When the probe came into contact with the cell surface (inset) and indented the cell up to a certain depth (depicted on *x*-axis), cantilever bending was measured by the optical fiber. The corresponding loading force is depicted on the *y*-axis. The measurements separately detected the loading (light blue), holding (orange), and unloading (green) periods. The fitted loading curve (red) was used to calculate the effective Young’s modulus using a Hertzian contact model. (**C**) The bar plot shows the effective Young’s moduli E_eff_ of Ishikawa cells grown on polyacrylamide gels of 1, 4, and 23 kPa stiffness. Cells were untreated or treated with estradiol (E2) or a combination of estradiol and medroxy-progesterone acetate (E2MPA) for 5 days before analysis. No changes in apical cell stiffness were observed with respect to substrate stiffness and hormonal treatment (mean ± SD, data obtained by at least 3 independent experiments with 3–9 measurements per experimental condition). (**D**) The bar plot depicts the average time in seconds that it took the cells to lower their intrinsic resistance toward the applied deformation to 70% of the initial force of resistance. Note that stress–relaxation was influenced by matrix stiffness. It was significantly slower in Ishikawa cells growing on 1 kPa substrates than on stiffer substrates in the presence of either estradiol or the combined presence of estradiol and medroxyprogesterone acetate (mean ± SD, data obtained by at least 3 independent experiments with 3–7 measurements per experimental condition).

**Figure 3 ijms-25-03726-f003:**
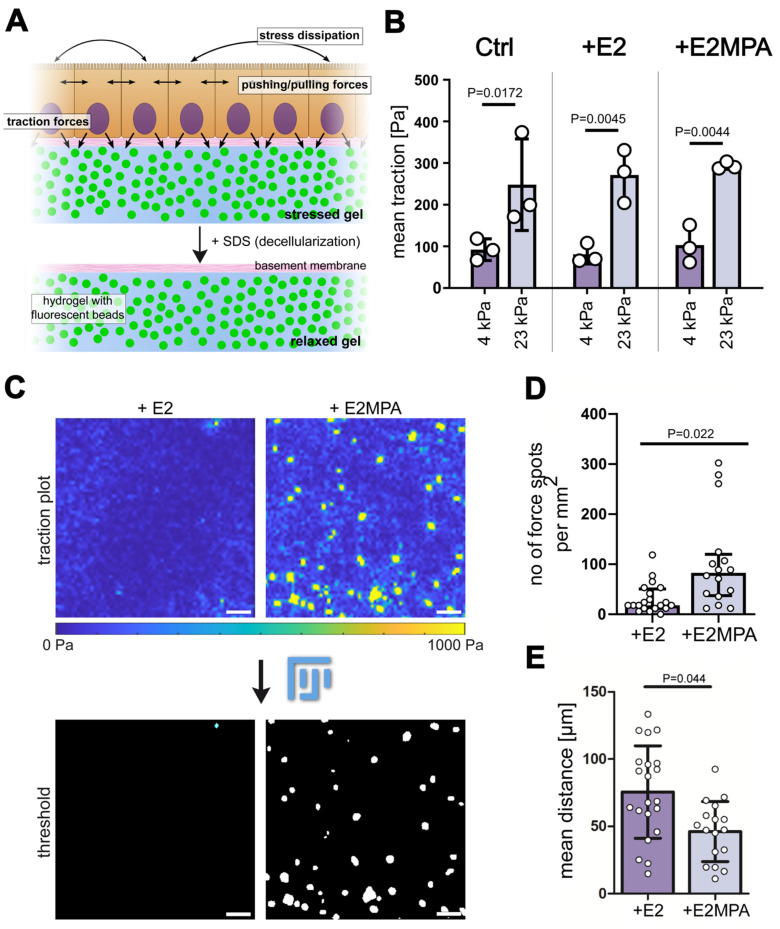
Steroid hormones affect the traction force patterns of Ishikawa monolayers. (**A**) The scheme depicts the principle of traction force microscopy (TFM). Cells were seeded on top of polyacrylamide (PAA) gels that contained fluorescent beads and an image of this stressed gel was obtained. After detaching the cells with SDS, the bead position in the relaxed gel was imaged. Cell-substrate stresses (tractions) could be calculated from bead displacement. (**B**) The bar graph shows the mean traction forces of Ishikawa monolayers grown on top of 4 kPa and 23 kPa polyacrylamide gels without hormone treatment or treatment with estradiol (E2) or a combination of estradiol and medroxyprogesterone acetate (E2MPA) for 5 days before analysis (data obtained by 3 independent experiments with at least 6 measurements per experimental condition; mean ± SD, one-way ANOVA). (**C**) Traction force heat maps of Ishikawa monolayers after hormone treatment. Scale bars = 50 µm. (**D**,**E**) The plots show the total number of force hotspots per mm^2^ and the mean Euclidian distance between hotspots in Ishikawa cells after steroid hormone treatment. Error bars represent the interquartile range (**D**) or standard deviation (**E**). Data were obtained in at least 3 independent experiments with 3 technical replicates per condition; Mann–Whitney U test (**D**) or unpaired *t*-test (**E**).

**Figure 4 ijms-25-03726-f004:**
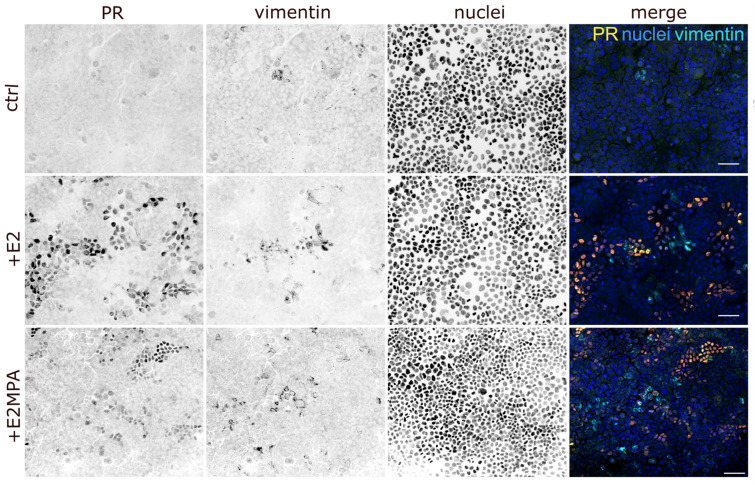
Progesterone receptors and vimentin are restricted to single cells and small cell clusters in Ishikawa monolayers. The fluorescence images of Ishikawa monolayers show the distribution of the progesterone receptor (PR; orange), vimentin (cyan), and nuclei (Hoechst 33342; blue). Monolayers were either untreated (ctrl) or stimulated with estradiol (E2) or a combination of estradiol and medroxyprogesterone acetate (E2MPA) for 5 days. The confocal images are displayed as maximum intensity projections (Z-step interval = 1.5 µm). Scale bars = 25 µm.

**Figure 5 ijms-25-03726-f005:**
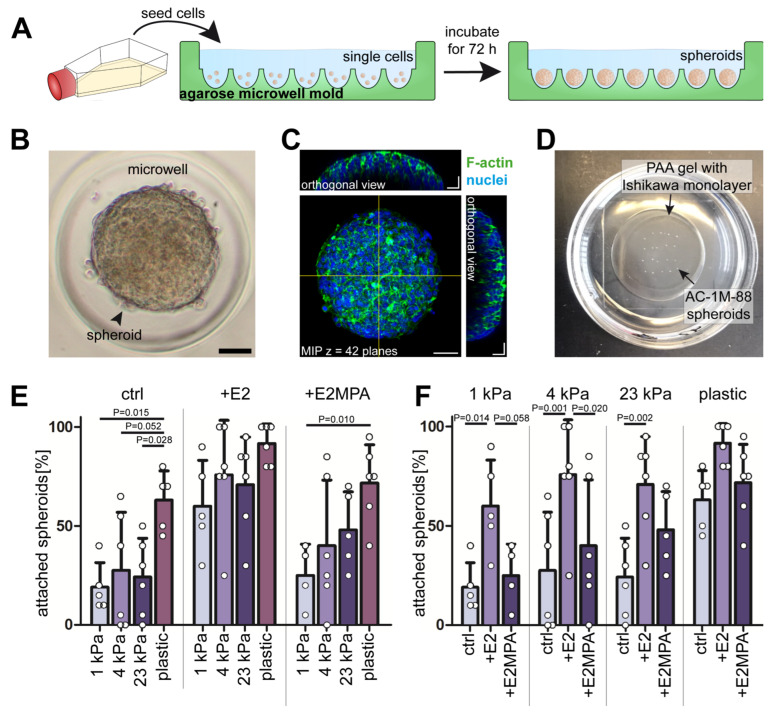
The trophoblast adhesion of Ishikawa monolayers depends on substrate stiffness. (**A**) The scheme depicts how trophoblast spheroids were prepared in scaffold-free agarose microwell molds. Agarose molds were custom-made by pouring 3.5% agarose onto a green-colored silicone mold. After sterilization by UV light, AC-1M-88 cells (~780 cells/microwell) were seeded into the agarose microwell molds, where they aggregated and formed multicellular spheroids after 72 h of incubation. (**B**) Representative bright field image of AC-1M-88 trophoblast spheroid in an agarose microwell. Scale bar = 50 µm. (**C**) Fluorescence image showing phalloidin-labeled F-actin (green) and Hoechst 33342-stained nuclei (blue) in a trophoblast spheroid. Corresponding orthogonal projections show the cross-section of the spheroid, as indicated by the yellow lines. Images are displayed as maximum intensity projection (MIP) with Z-step intervals of 1.5 µm. Scale bars = 50 µm (MIP) and 25 µm (orthogonal view). (**D**) The photograph shows trophoblast spheroids that had been placed manually on top of an Ishikawa monolayer at regular intervals using an embryo transfer pipettor, without moving the dish to avoid displacement of the spheroids. After 2 h of incubation, cells were washed 3X with PBS and the remaining spheroids were counted. PAA (polyacrylamide). (**E**,**F**) The bar plots show the percentage of attached spheroids on top of Ishikawa monolayers after 2 h of co-culture and consecutive washing. The same data set was used to highlight either stiffness dependence (**E**) or hormone dependence (**F**). Ishikawa monolayers were seeded either on soft PAA gels (1 kPa, 4 kPa, or 23 kPa stiffness) or on plastic and either not hormone treated or treated with either estradiol alone (E2) or a combination of estradiol and medroxyprogesterone acetate (E2MPA) for 5 days prior to the confrontation assay (mean ± SD, data obtained by at least 4 independent experiments; two-way ANOVA with Bonferroni post-hoc text).

## Data Availability

The data underlying this article are available in the article and in its online Appendix A.

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
