# Peer review of "An Assessment of the Mechanophysical and Hormonal Impact on Human Endometrial Epithelium Mechanics and Receptivity"

_ijms, 2024, doi:10.3390/ijms25073726_

Round 1
Reviewer 1 Report
Comments and Suggestions for Authors
11. Lines 417-419: “Final concentrations were 1×10–8 M for E2 and 1×10–6 M for MPA. Hormone diluent ethanol was added to the medium of the unstimulated control (0.03% v/v). Monolayers were cultivated for 5 days.” Please, describe how did you measured the final concentration of the hormones in the medium? Or there are given calculated concentrations? Discuss the fact that hormone levels fall due to the degradation during cell culture and how it can influence on the results.
22. Line 423 “by pouring 3% agarose dissolved in H2O (w/v, Biozym Scientific GmbH” Which kind of H2O was used: distilled, deionized?
33. Chapter 5.2. May be it will be good to provide some graphic schemes describing the setup of experiments.
44. Lines 441-442: “After 2 hours, monolayers were washed 3X with PBS and the number of attached spheroids was counted.” Which volume of PBS was used for washing? What was direction of washing? Probably, in some cases washing jet acted directly on spheroids and removed them?
55. Line 487 “Cells were treated with either E2 or the combination of E2 and P4 for 5 days”, lines 415-417 “Cells were treated with either 17β-estradiol (E2) or the combination of 17β-estradiol and medroxyprogesterone acetate (MPA, Sigma-Aldrich)”. What did you used: P4 or MPA?
66. Discuss the selected concentration of used hormones and duration (5 days) of experiment.
77. Lines 565-566: “Parametric data is presented as mean ± standard deviation (SD), non-parametric
as mean ± 95% confidence interval.” It is better to show non-parametric data as median, interquartile range and variation. Also, please, indicate in which cases you observed non-parametric distributions?
88. Figure 5E. What is the stiffness of plastic? What kind of plastic did you used? Provide some pictures with Ishikawa cells grown on plastic. There is given only example for glass (Figure 1B).
99. Lines 350-351 check text formatting.
110. Lines 351-352. Discuss in more details the experimental differences of you work and [39]: time of hormone treatments of Ishikawa cells, differences between mouse blastocysts and your spheroids, menstrual and endometrial cycles regulations, anchoring proteins in both cases.
Reviewer 2 Report
Comments and Suggestions for Authors
The manuscript titled “Assessment of the mechanophysical and hormonal impact on human endometrial epithelium mechanics and receptivity” by Sternberg, A.K.; et al. is a scientific work where the authors study the mechanical cues that affects to the underlying mechanisms of embryo implantation to endometrium at the single molecule level. For it, the authors used a combination of fluorescence microscopy and nanoindentation measurements. This is a well-designed work and it could be interesting for a certain target audience specialized in this field. Furthermore, the manuscript is generally well-written.
However, it exists some points that need to be addressed (please, see them below detailed point-by-point). The most relevant outcomes remarked by the authors can contribute in the growth of many fields like the healthcare in order to design the next-generation of strategies to ameliorate the pregnancy rates.
. For this reason, I will recommend the present scientific manuscript for further publication in the International Journal of the Molecular Sciences once all the below described suggestions will be properly fixed.
Here, there exists some points that must be covered in order to improve the scientific quality of the manuscript paper:
1) INTRODUCTION. This section clearly depicts the current state-of-the-art of the examined biological systems but it lacks brief statement about the significance of the used nanoindentation technique [1] to ascertain the mechanical cues of living cells and how they can affect to mechanotransduction processes [2].
[1] Magazzù, A.; et al. Investigation of Soft Matter Nanomechanics by Atomic Force Microscopy and Optical Tweezers: A Comprehensive Review. Nanomaterials 2023 13, 963. https://doi.org/10.3390/nano13060963.
[2] Andreu, I.; et al. The force loading rate drives cell mechanosensing through both reinforcement and cytoskeletal softening. Nat. Commun. 2021, 12, 4229. https://doi.org/10.1038/s41467-021-24383-3.
2) RESULTS. “Given the endometrial (…) on polyacrylamide-based hydrogels with defined stiffness” (lines 74-77). Here, the authors used different crosslinking bis-acrylamide to obtain these surface materials with different stiffnesses. Did the authors check this parameter for all the hydrogels prior the cell seeding? Some information was already detailed in the Fig. S2 (Supplementary Information) but it is not clear if this test was done by triplicate or for all the formed hydrogels prior its use. Why the polyacrilamide gel of 4 kPa exhibits larger data scattering in comparison to the analogous of 1 kPa and 23 kPa, respectively?
3) “In the next set of experiments, we therefore studied the mechanical properties of Ishikawa cell monolayers growing on hydrogels with different stiffness by nanoindentation” (lines 134-136). Did the authors observe a cellular dettachment during the nanoindentation measurements?
4) Figure 2, panel B (line 146). Did the authors conduct some baseline correction to the gathered force-distance curves? Some information should be provided in this regard.
5) “From that, the forces exerted (…) at high spatial resolution” (lines 190-191). The estimated radius of the sphere beads ranges from 8.5 to 10.0 µm. Do the authors consider that with this type of probes high lateral resolution could be achieved?
6) Subsection 2.3. “4 kPa and 23 kPa hydrogels were compared as physiological and supraphysiological stiffness” (lines 191-192). Did the authors test the impact of a very stiff surface like glass (in order of GPa) in this set of experiments? Some information should be furnished in this regard.
7) DISCUSSION. This section clearly states the most relevant outcomes found in this work. No actions are requested to the authors.
8) CONCLUSIONS. Here, it may be convenient if the authors could add a brief statement about the future action lines to pursue this research.
9) MATERIALS & METHODS. This section provides many information details to mimic the devoted experiments with accuracy. One of the main cornerstones of this research is the nanoindentation measurements (lines 519-541). How did the authors calibrate the AFM lever? Did the authors qualify the bead radius before and after the nanoindentation measurements? Did the authors observe changes in the tip radious morphology during the devoted nanoindentation measurements (e.g. by SEM images)? Some information should be provided in this regard.
Comments on the Quality of English LanguageThe manuscript is generally well-written albeit a final revision by the authors may be advisable in order to polish final details.
Round 2
Reviewer 1 Report
Comments and Suggestions for Authors
Can be accepted.
Reviewer 2 Report
Comments and Suggestions for Authors
The authors did a great deal of effort to cover all the suggestions raised by the Reviewers. For it, the scientific quality of the manuscript was greatly improved. Based on the sugnificance of this research and scope of IJMS, I warmly endorse this work for further publication in IJMS.